# Energy and Key Micronutrient Intake in Amateur Swimmers: A Pilot Study

**DOI:** 10.3390/nu17040664

**Published:** 2025-02-13

**Authors:** Sara Gonçalves, Filipa Vicente, Paula Pereira

**Affiliations:** Applied Nutrition Research Group (GENA), Nutrition Lab—Egas Moniz Center for Interdisciplinary Research (CiiEM), Egas Moniz School of Health & Science, Campus Universitário, Quinta da Granja, 2829-511 Caparica, Almada, Portugal; sara.nutricao22@gmail.com (S.G.); pmpereira@egasmoniz.edu.pt (P.P.)

**Keywords:** swimming, food log, diet intake, calcium, zinc

## Abstract

Swimming is a highly demanding sport that requires the integration of multiple physical, physiological, and psychological factors for optimal performance. Despite its popularity, most swimmers are amateur, and few studies have assessed energy and nutrient intake at this athlete level. Therefore, the aim of the present pilot study was to evaluate energy and nutrient intake and compare them with the recommended values. **Methods**: The participants were recruited from local teams. To determine body composition, weight, height, and skinfold measurements were taken. Food intake assessment was conducted through a 3-day food log. **Results**: The sample was composed of 15 individuals, where 7 were female. There were no significant differences in energy and macronutrient intake between male and female swimmers, nor between rest or workout days. Despite protein intake being within the recommendation, the estimated average intake for energy was significantly lower than the recommended values. Female swimmers also showed an inadequate intake of calcium, iron, and zinc. **Conclusions**: Swimmers showed an inadequate energy and carbohydrate intake for their activity level. Female athletes also reported a low intake of crucial micronutrients.

## 1. Introduction

Swimming has been a popular Olympic sport since 1896, with distances ranging from 50 to 1500 m and various swimming styles (crawl, breaststroke, butterfly, and back swimming) leading to competitive races lasting 20 s to 15 min [1].

This sport is highly demanding, requiring the integration of multiple physical, physiological, and psychological factors for optimal performance. Unlike many other sports, swimming engages nearly all major muscular groups simultaneously, necessitating both strength and endurance [2,3,4]. The aquatic environment adds complexity by requiring athletes to overcome water resistance, which raises the energy cost of movement [5]. Additionally, swimmers must develop efficient breathing techniques, as their access to oxygen is limited by the rhythmic nature of the stroke cycle. Physiologically, swimming places significant demands on the cardiovascular and respiratory systems, requiring a high level of aerobic capacity and anaerobic power. Moreover, the precision and coordination needed for technical execution (e.g., stroke mechanics) demand fine motor control and mental focus [6,7,8].

Swimmers’ career often begins during early ages, frequently in childhood, and reaches peak performance when they are young adults aged 22–24 years old [9]. Most of these swimmers do not reach the peak performance defined as elite and therefore keep a competitive, but amateur, level of practice despite large training volumes and high intensity, remaining close to elite level [10,11,12].

Therefore, swimming characteristics as a sport lead to the importance of an appropriate nutritional status and optimal body composition. An adequate body fat percentage reduces drag in the water while lean body mass is associated with strength and power and therefore performance [13,14,15]. The high volume of training and the considerable intensities justify the high energy and macronutrient demands; therefore, swimmers must maintain an appropriate diet to meet carbohydrate and energy requirements. Because of the length and intensity of training sessions, swimmers’ glycogen stores can be drained in a few sessions, and given that they exercise every day, sometimes twice a day, proper carbohydrate intake is mandatory [1]. These facts highlight the importance of diet considering that food intake must meet the recommendations according to the season-specific phase [1,16].

Despite the fact that swimming is a very well-known sport and is one of the most frequently practiced in Europe, there is no clear and robust evidence of whether food and nutrient intake in swimmers is in accordance with recommendations and requirements. Very few studies have addressed this subject and data are quite heterogenous in terms of age groups, food intake assessment methods, and body composition analysis [17,18,19].

In addition to macronutrient-specific recommendations, sports nutrition guidelines have referenced some micronutrients of concern, namely vitamin D, calcium, iron, and zinc [20,21]. Jordan and colleagues also reported that it is common to have a low intake of these micronutrients [22]. Except for vitamin D, the other micronutrient status that has been referenced is primarily determined by dietary habits, emphasizing the importance of robust data on their intake, which remains a research gap especially at the amateur level.

Additionally, no known studies have performed a qualitative analysis of food intake, namely the characterization of dietary patterns. The Mediterranean diet is one of the most frequently recommended dietary patterns in Europe, and evidence has shown multiple benefits in human health and chronic disease prevention [23,24]. More recently, it has also been recommended as a healthy diet model for sports practice [25,26]. Evidence on athlete adherence is scarce, including in Mediterranean countries like Portugal [27] and Spain [28], which also represents a research gap for further studies.

Therefore, the aim of the present study was to evaluate the food intake of amateur swimmers and compare them with the recommendations for energy and crucial micronutrients for this age group, as well as the adherence to a healthy food pattern, the Mediterranean diet.

## 2. Materials and Methods

### 2.1. Sample and Ethical Aspects

This cross-sectional study was carried out with a sample of amateur athletes. The individuals who joined the sample were briefed on the objectives and methods of the study, and those who gave their informed consent moved forward with the research. The study received approval from the Egas Moniz Ethical Committee, and all procedures complied with the Helsinki Declaration for human studies.

The athletes were recruited by announcement in the local teams, and those who were swimming at least three times per week and had competed within the previous six months were included. The individuals who reported fewer than three weekly training sessions, had never competed, or did not attend the body weight and body composition assessments were excluded.

### 2.2. Anthropometric and Body Composition Assessment

All anthropometric procedures were performed in a private room with a temperature of 20 °C. Participants were asked about their birth year, nationality, training frequency, other exercise practice (e.g., sports, frequency), and most recent competition data (e.g., swimming style, distances).

A wall-mounted precision stadiometer (SECA) was utilized for height measurement with millimeter accuracy. Participants kept their shoulder blades, back of the head, and buttocks against the board while keeping the head in Frankfurt’s horizontal plane. Weight was measured to an accuracy of 100 g using a Tanita BC-543 scale; both measurements were taken after a complete expiratory movement.

The biceps, triceps, subscapular, and supra iliac skinfolds were measured with an INNOVACARE Cescorf caliper. The anthropometric assessment also included arm, waist, thigh, and calf circumferences. All these measurements were conducted according to the ISAK procedures by a level 1 technician [29]. Female swimmers’ body density was evaluated using the Durnin and Wommersley [30] formula, whereas male participants’ body density was estimated using the average of the Ritters and Reilly–Wallace formulas [31,32]. The Siri equation was used to calculate body fat. Muscle mass was estimated according to the Lee formula [33].

### 2.3. Food Intake Assessment

The food intake was assessed using a three-day food diary. Participants were instructed to report all foods and beverages consumed, including the time, place, and preparation details. This three-day food record included 2 workout days and 1 rest day, to be representative of a common training week. A typical illustrated model book containing three different portion sizes for the most frequent dishes consumed in Portugal aided with portion size estimation. In addition, swimmers completed the 14-Item Mediterranean Diet Adherence Screener (MEDAS) to assess adherence to the Mediterranean diet pattern [34].

Data from the three-day food record were introduced into a datasheet and computed to determine total energy and macronutrient intake, as well as calcium, iron, and zinc intake. A Portuguese-specific food composition table and a European table of homogeneous, multi-ingredient foods were used [35,36].

The energy needs were estimated by first calculating each swimmer’s basal metabolic rate calculated using the Harris–Benedict formula and adding the adequate physical activity factor according to the reported lifestyle data considering the weekly frequency of training sessions and the estimated duration as well as the daily routine described in the initial assessment. This procedure follows the recommendations for estimating energy requirements in European adults [37].

### 2.4. Data Analysis

All statistical analyses were carried out using SPSS version 29 (IBM, Armonk, NY, USA). For categorical data, descriptive results are presented as the number of participants (%), and for Gaussian and non-Gaussian distributed continuous variables, they are presented as the mean ± SD.

The normality of continuous variables was assessed using the Shapiro–Wilk test. The Friedman test was employed to compare energy and macronutrient consumption on training days versus non-training days, as well as to assess intake against recommendations. A one-sample *t*-test was employed to compare micronutrient intakes against dietary reference intakes. A two-sided test is considered statistically significant when *p* < 0.05.

## 3. Results

The overall recruited sample consisted of 20 individuals, only 17 gave their informed consent, and 15 completed all the procedures included in this study.

### 3.1. Anthropometric and Body Composition Data

Most of the participants had a normal weight and appropriate body fat levels (Table 1). The average BMI was 23.0 ± 2.6, which is within the interval for normal weight [38], and the average body fat in female swimmers was 24.3% ± 6.0 and that in male swimmers was 10.9% ± 3.9, which can be considered “normal/average” values [39].

One female participant had a higher body fat mass than was advised, while two had values under the recommended interval. The male participants’ body mass index demonstrated that they were all of normal weight according to the body mass index classification [38], despite the fact that two male swimmers had low body fat levels [39]. As expected, men swimmers had a larger average estimated muscle mass than female swimmers.

### 3.2. Energy and Macronutrient Intake

The analysis of data reported in the food diaries is presented in Table 2; there were no significant differences in energy and macronutrient intake between male and female swimmers (*p* < 0.05). The participants showed a lower energy intake on the rest days when compared to workout days, but it was not significantly different (*p* > 0.05). However, on both workout and rest days, the estimated average energy intake was significantly lower than recommended for these female and male participants (*p* < 0.05). Female swimmers had a higher protein intake on rest days, while male swimmers had a higher protein intake on workouts days, but this was not significantly different (*p* > 0.05). Globally, swimmers had higher protein and carbohydrate intakes on workout days.

The macronutrient contribution to total energy intake was within the acceptable macronutrient dietary range (Appendix A). The average intake per kg is presented in Table 3; there were no differences between protein and carbohydrate intakes when comparing workout and non-workout days, as well as between female and male swimmers. The average protein intake in this sample was within the recommended values for sports practice, but the same was not shown in carbohydrate intake despite the fact that there were no differences [20].

### 3.3. Data on Calcium, Iron, and Zinc Intake

Table 4 presents the average intake of key micronutrients in sports nutrition [20,21]. The average calcium, iron, and zinc intake in female swimmers was significantly lower than the average requirements in this age group (*p* < 0.05) [37]. In male participants, average calcium and zinc intakes were lower than the requirements but not significantly different.

### 3.4. Adherence to Mediterranean Diet Pattern

The average score in the MEDAS questionnaire was 7.5 ± 2.8, which is considered moderate to fair adherence [34]. Within the participants, 6 showed a high adherence to the Mediterranean dietary pattern, while 5 participants showed a low adherence (Table 5).

A more detailed analysis concerning each question showed that few athletes reported daily nut consumption (n = 4) and included pulses in their diet (n = 6). As expected, no athlete reported daily wine consumption. Most athletes had at least 3 fruit servings (n = 11), while 9 had at least 3 daily portions of vegetables.

## 4. Discussion

As previously referred, there are few studies concerning the food and nutrient intake of amateur athletes. In the present work, the results revealed an inadequate intake of calcium, iron, and zinc in female swimmers. In addition, one-third showed a low adherence to the Mediterranean dietary pattern.

### 4.1. Body Composition Assessment

The body composition analysis revealed that most swimmers had normal body fat and body weight values, as expected in physically active individuals. Despite this, the average body fat in female athletes was higher than that observed in previous studies [1,40,41,42]. This could be expected considering that they compete at an amateur level only in regional and national events. Previous research has shown that body composition, particularly an adequate level of body fat, influences swimming performance [13,40]. It is known that this body composition can vary throughout the season [43]. The present study was conducted in an intermediate stage to ensure that it was representative of an intensive training period leading up to a competitive period.

### 4.2. Energy and Macronutrient Intake

Nutritional assessment was performed using a 3-day food diary, which revealed a low energy intake compared to energy requirements in this sample. This has been noted in other sports at the amateur level [44,45], although some data have shown that energy intake is adequate [46,47].

The method used to assess dietary intake has a major influence on these results; several studies have used food frequency questionnaires, whereas in the present paper, the food diary was the method of choice. While food frequency questionnaires are based on participants recalling and reporting the frequency and portion sizes of foods consumed over a longer period, typically weeks to months, this method provides a broad overview of habitual intake but is prone to recall bias and may under- or overestimate energy intake due to inaccurate recall or portion size estimation [48,49].

On the other hand, food diaries involve the real-time recording of everything consumed over a shorter, specific period, usually several days. This approach can provide more accurate data as it records immediate intake, reducing reliance on memory. However, food diaries can be affected by under-reporting, especially in people who want to appear healthier, and they can change participants’ eating behavior due to the burden of constant recording [50,51]. This may also lead to lower adherence among participants, which justifies the sample size in this study.

The average carbohydrate intake was also below the recommended values for elite athletes, as shown in other studies, including some conducted in swimmers [17,19,52]. In sports nutrition guidelines, it is advised that athletes have 5 to 7 g of carbohydrates per kg of weight daily when training for 1 h every day [20].

Popular dietary trends, particularly those associated with weight control, as well as an excessive focus on protein intake for muscle growth/maintenance, justify the tendency of some athletes to reduce their carbohydrate intake [53]. Nutritional knowledge and literacy influence athletes’ habits leading to these practices [54], and social media and peers are known to influence their beliefs about nutrition and diet [55]. In spite of this, it is important to consider that carbohydrate intake can be manipulated according to training load and season stage [1] and range from 3 to 10 g/kg/day. Nevertheless, considering that the team members were not receiving nutritional support and this study was conducted in an intermediate phase of the competitive season, the carbohydrate intake could be considered below the recommendations.

### 4.3. Micronutrient Intake

In the present study, the female swimmers also showed an inadequate intake of calcium, which has previously been shown in swimmers [18,46] but also in other sports [56,57,58]. Athletes tend to avoid milk or even all dairy products and do not have other sources in sufficient quantities. For example, few participants in the present review included the daily consumption of nuts, which could be an option to increase intake, such as almonds. These data are of concern given the importance of calcium for bone health and homeostasis [59,60,61]. There is a growing trend towards avoiding milk and preferring non-dairy alternatives (e.g., soy drinks, almond drinks). In the past, this was due to lactose intolerance or maldigestion, or even cow’s milk allergy, which necessitated the complete absence of milk in people’s diets, and this was a possible risk factor for low bone mineral density [62]. Recently, several other motives for avoiding these products have been suggested, some related to the environmental impact of animal food sources, others to food preferences, and there are also misconceptions about the effects of milk consumption on health. This remains a controversial issue, but it is important to consider the considerable heterogeneity in the nutritional composition of these vegetable drinks, and not all are fortified with vitamin D, calcium, or other micronutrients of interest that could be provided by dairy products, leading to multiple nutrient deficiencies [63,64]. In the present study, several participants consumed only one portion of dairy products per day. It is also of concern whether these athletes intend to use supplements to replace dietary sources, given the negative effects of calcium supplementation reported in some studies [65,66].

These results also showed that zinc intake was also below recommendations in female swimmers. It is estimated that 17% of the global population has an inadequate zinc intake [67,68]. This trace element is generally considered a gatekeeper in immune function [69]; therefore, an inadequate intake is thought to be a risk factor for an increased risk of infection. Specifically, it has been shown that competitive athletes are quite prone to upper respiratory tract infections [70], including swimmers [11]. An appropriate zinc intake could have some influence on respiratory health outcomes [71,72]; therefore, this is also a matter of concern especially because there is some evidence of low levels of zinc body biomarkers even at higher intakes [73]. In the present sample, the low frequency of nut consumption could justify this reduced zinc intake [74]. This is not a novelty in Portuguese athletes; a study from Nunes et al. [75] also showed an inadequate intake of this trace element even in highly trained athletes.

### 4.4. Mediterranean Dietary Pattern Adherence

The results have also shown that only 40% of the swimmers presented a high adherence score to the Mediterranean dietary pattern. Despite being a recent topic in sports nutrition, several benefits have been attributed to the Mediterranean diet in sports outcomes [26]. The meta-analysis conducted by Bizzozero-Peroni et al. [76] showed that in spite of the high adherence to a Mediterranean diet, it was associated with higher levels of cardiorespiratory and musculoskeletal fitness and overall physical fitness in adults. In the present sample, participants reported no daily wine intake and very few had nuts or pulses daily. Regarding pulses, a previous study conducted in Portugal [77] suggested several barriers to their inclusion in the daily diet, namely the possible gastrointestinal effects (e.g., abdominal cramps, flatulence) that could interfere with athletes’ training sessions. Nevertheless, athletes can be better informed about its human health benefits and cooking techniques to reduce secondary effects (e.g., timing).

No similar studies are known to have been carried out in relation to nuts, suggesting that the possible fear of their energetic value and its influence on body weight and fat mass could lead to a lower consumption [78].

The assessment of adherence to the Mediterranean diet is controversial among athletes. They tend to show higher adherence than the general population, but there are significant differences between different sports and age groups, which could be due to the different questionnaires and other methods that can be used [79]. As shown in the present study, the two questions on the daily consumption of wine and pulses may reduce the score of athletes, and there are specific constraints on including these foods in their daily diet.

### 4.5. Strengths and Limitations

There is a lack of studies on food intake assessment and nutrient adequacy in amateur athletes, especially in Portugal; therefore, this study adds up the data on this subject. It is also important to highlight the gold-standard methods used in this study that can be replicated in the field, namely body composition assessment through skinfolds and food records as food intake assessment methods.

Despite this, our results should be interpreted with caution due to the small sample and, therefore, this pilot study can justify the need for larger studies. Although food records are valid tools for food intake assessment in this context, there are several biases involved that can lead to the underestimation of intake due to food omission. Adherence can also be affected due to the participant burden, and this justifies the sample size.

## 5. Conclusions

In this pilot study, athletes showed a lower energy intake than recommended for their age, weight, and activity level, which may affect their performance, but also their health and well-being in the long term. Female swimmers were also found to have a worryingly low intake of calcium and zinc. There is also a need for further studies to add more data on food intake on these athletes, who are often unattended to.

The results also reinforce the need for competitive athletes to have adequate nutritional support in their clubs, even at lower league and amateur levels.

## Figures and Tables

**Table 1 nutrients-17-00664-t001:** Sample anthropometric and body composition data analysis.

	Female	Male	Total
**N**	7	8	15
**Age**	23 ± 4	22 ± 3	23 ± 3
**BMI**	22.9 ± 3.3	23.0 ± 2.0	23.0 ± 2.6
Normal weight (n/%)	6/85.7%	8/100%	14/93.3%
Overweight (n/%)	1/14.3%	--	1/6.7%
**% Body fat**	24.3% ± 6.0	10.9% ± 3.9	n/a
Low (n/%)	2/28.6%	2/25.0%	4/26.7%
Normal (n/%)	4/57.1%	6/75.0%	10/66.7%
Excessive (n/%)	1/14.3%	--	1/6.7%
**% Muscle mass**	36.7% ± 6.1%	43.9% ± 6.5%	n/a

**Table 2 nutrients-17-00664-t002:** Energy and macronutrient intake in female and male swimmers on rest days and workout days.

	Female	*p*-Value	Male	*p*-Value	Total	*p*-Value
**Energy W (kcal)**	2195 ± 428	0.500	2093 ± 669	0.273	2131 ± 582	0.767
**Energy R (kcal)**	1871 ± 504		1990 ± 540		1941 ± 505	
**Protein W (g)**	104 ± 19	0.223	117 ± 34	0.465	112 ± 29	0.477
**Protein R (g)**	110 ± 34		120 ± 37		107 ± 33	
**Carbs W (g)**	282 ± 50	0.500	241 ± 106	0.273	256 ± 90	0.906
**Carbs R (g)**	239 ± 57		228 ± 60		233 ± 57	
**Fat P (g)**	67 ± 32	0.786	68 ± 28	0.715	67 ± 29	0.767
**Fat R (g)**	48 ± 25		69 ± 30		60 ± 29	

W = workout days; R = rest days.

**Table 3 nutrients-17-00664-t003:** Average carbohydrate and protein intakes per kg on rest and workout days.

	Female	Male	Total
**Workout days**			
Protein (g/Kg)	1.7 ± 0.4	1.8 ± 0.5	1.7 ± 0.5
Carbohydrates (g/Kg)	4.0 ± 1.0	4.1 ± 1.1	4.1 ± 1.1
**Rest days**			
Protein (g/Kg)	1.5 ± 0.5	1.7 ± 0.4	1.6 ± 0.4
Carbohydrates (g/Kg)	4.1 ± 0.6	2.7 ± 2.1	3.9 ± 0.9
**Workout and rest days**			
Protein (g/Kg)	1.5 ± 0.1	1.8 ± 0.1	1.7 ± 0.2
Carbohydrates (g/Kg)	4.1 ± 0.4	3.6 ± 1.1	4 ± 0.9

**Table 4 nutrients-17-00664-t004:** Average calcium, iron, and zinc intakes in female and male swimmers. The *p*-value refers to the comparison with the average requirements considered in the data analysis [37].

	Female	*p*-Value	Male	*p*-Value	Total
Calcium	445.2 ± 113.1	0.028	637.10 ± 353.7	0.060	548.5 ± 278.9
Iron	8.2 ± 2.3	0.020	10.79 ± 5.9	0.060	9.6 ± 4.7
Zinc	7.2 ± 1.9	0.040	10.1 ± 5.9	0.116	8.7 ± 4.6

**Table 5 nutrients-17-00664-t005:** Sample distribution based on adherence to the Mediterranean diet according to results from MEDAS [34].

	n/%
Low adherence	5/33%
Moderate adherence	4/27%
High adherence	6/40%

## Data Availability

The original contributions presented in this study are included in the article/Appendix A. Further inquiries can be directed to the corresponding author.

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
