# Peer review of "Energy and Key Micronutrient Intake in Amateur Swimmers: A Pilot Study"

_nutrients, 2025, doi:10.3390/nu17040664_

Round 1
Reviewer 1 Report
Comments and Suggestions for Authors
Thank you for submitting your manuscript which is unfortunately lacking some major technical issues that need to be solved before a proper review can be completed.
Your manuscript requires thorough language editing. Please have a native speaker or proper software check it. And correct it accordingly.
You fail to use consistent referencing. Please correct this.
Your abstract refers to youth athletes while the participants are adults.
While your title refers to „Amateur Swimmers“ your Materials & Methods talk of „Sub-Elite Swimmers“.
Your title refers to „zinc and calcium“ although your have calculated iron well.
Please rethink your title and these issues for better understanding.
An introduction is there to lead to the scientific question. However, it is not clear from your introduction why the micronutrients zinc, calcium, iron or the Mediterranean diet are of interest to athletes. Please rewrite your introduction.
Have you calculated energy expenditure in any of the swimmers? This data needs to be presented in order to state energy deficiency. Have you considered that there might be a difference in energy expenditure in different swimming disciplines and training intensity?
The reviewer doubts that by collecting diet data for three (!!) days, it is possible to compare training and competition days. Please explain your reasoning and underline this with statistical sound arguments.
The results section is meant to refer to your numerical results and to be presented as such.
You, however use this already for discussion and interpretation of your data; such as „normal weight“ and „appropriate body fat“ without any scientific reference. Please rewrite this for clarity.
Line 130: If there is no significant difference, this means there is no difference. Please rewrite this sentence.
We look forward to reading your amendments and new manuscript.

Your manuscript requires thorough language editing. Please have a native speaker or proper software check it. And correct it accordingly.
Author Response
The authors would like to thank you for taking the time to review this manuscript and for the valuable recommendations you had given to this work. Please find the detailed responses below and the corresponding corrections highlighted in the new version.
Reviewer 1
Your manuscript requires thorough language editing. Please have a native speaker or proper software check it. And correct it accordingly.
The authors acknowledge the comment of the reviewer. The work was edited and proofread for grammar errors.
You fail to use consistent referencing. Please correct this.
Your abstract refers to youth athletes while the participants are adults.
Thank you for pointing this out. This was corrected (line 11).
While your title refers to „Amateur Swimmers“ your Materials & Methods talk of „Sub-Elite Swimmers“.
Thank you for pointing this out. In order to keep consistency, we changed to amateur athletes (line 52).
Your title refers to „zinc and calcium“ although your have calculated iron well.
Please rethink your title and these issues for better understanding.
We agree with the comment therefore we changed the title, considering that only female had inadequate intakes we removed the word “low”.
An introduction is there to lead to the scientific question. However, it is not clear from your introduction why the micronutrients zinc, calcium, iron or the Mediterranean diet are of interest to athletes. Please rewrite your introduction.
The authors agree with the comment and revised the introduction, adding an important state of art that leads to the study research questions and aim.
Have you calculated energy expenditure in any of the swimmers? This data needs to be presented in order to state energy deficiency. Have you considered that there might be a difference in energy expenditure in different swimming disciplines and training intensity?
We agree with the comment and we added a comprehensive description of this procedure (lines 92-94). The use of Accelerometers in swimming is very scarce at this level due to economic reasons, but it would be the best option to monitor the energy expenditure during the workout sessions. Nevertheless, to our best information we estimated the energy expenditure based on training frequency, estimated duration of each session given by the coach and the athletes and the perceived effort sensations.
The reviewer doubts that by collecting diet data for three (!!) days, it is possible to compare training and competition days. Please explain your reasoning and underline this with statistical sound arguments.
We acknowledge your concerns. First, we would like to clarify that the food log only considered training or rest days, not competition days. We asked for a food record for 2 workout days and 1 rest day. When we analyzed the food records there were no differences in energy and nutrient intake therefore we considered the average intake, in spite we also reported the average for rest and workout days (table 2).
The choice for food records was taken considering that it is a gold standard method and the sample characteristics (adults, mostly normal weight, informed and motivated). The authors though about extending the 3-day period to 7 days but this could reduce the participant adherence as well as be a bias factor due to some impact in food intake or the report accuracy.
The results section is meant to refer to your numerical results and to be presented as such.
You, however use this already for discussion and interpretation of your data; such as „normal weight“ and „appropriate body fat“ without any scientific reference. Please rewrite this for clarity.
We acknowledge the comment. This interpretation was considered for classification purposes, we added the adequate references (20, 21).
Line 130: If there is no significant difference, this means there is no difference. Please rewrite this sentence.
We agree with the reviewer and correct this sentence.
Reviewer 2 Report
Comments and Suggestions for Authors
This is a small pilot clinical study with adequate novelty. However, some points should be addressed.
- The sample size is too small (15 swimmers) in order to derive reliable conclusions. Thus, the authors should add at the title that is a small pilot/preliminary study. This statement should also be added in the Abstract of the manuscript.
- The Introduction section is too small and it is uniquely-mainly referred to profesional and not amateur swimmers that take part in sports competitions where the energy intake and nutritional status are different from amateur swimmers. Thus, it is highly recommended to refer to amateur swimmers.
- At the end of the Introduction, the authors should report the aim of their study.
- Before the aim of the study, the authors should report the literature gap that exist in the international literature that their study will cover.
- The second and the third paragraph should be enriched with data concerning amateur swimmers.
- In line 52, the authors report that their study include sub-elite swimmers. This statement should be further described.
- The description of the sample in lines 57-61 should be reported initially reported in the Introduction section to explain the target study population of the article.
- A reference is required in line 92.
- In section 2.4, the normality test that examined the continious variables should be reported.
- In line 115, the p-value should be over 0.05.
- A more detailed description concerning the results of Tables 1, 2 and 3 should be performed, even if there are not significant differences.
- In Table 4, p-values should be added.
- Since the swimmers of the sample is athletes, the authors should report the exact type of swimming that they are specified. This should be also analysed since each type of swimming is associated with different energy intake.
- In section 4.2 the sentence in lines 168-171 is very complex and ton easily understood.
- The paragraph in lines 186-190 needs more analysis. The first sentence of this paragraph is complex and not easily understood.
- Based on the Methods-Results-Discussion section, the authors could report in the title of the paper that they studied amateur athletes swimmers.
- In section 4.3, the two last paragraph should be merged and should be enriched by further information.
- In the Discussion a separate section should be added by reporting the strengths and the limitations of the study.
- The main concern of the study is the too small sample size which can not support the reported conclusions in lines 256-261.
In the Conclusion section the authors should report what future studies could be performed in the future based on the results of their study.
Author Response
The authors would like to thank you for taking the time to review this manuscript and for the valuable recommendations you had given to this work. Please find the detailed responses below and the corresponding corrections highlighted in the new version.
- The sample size is too small (15 swimmers) in order to derive reliable conclusions. Thus, the authors should add at the title that is a small pilot/preliminary study. This statement should also be added in the Abstract of the manuscript.
The authors acknowledge the comment, added in the title, in the abstract and a comment in the conclusion. There is a clear lack of data on food and nutrient intake adequacy in amateur athletes in Portugal.
- The Introduction section is too small and it is uniquely-mainly referred to profesional and not amateur swimmers that take part in sports competitions where the energy intake and nutritional status are different from amateur swimmers. Thus, it is highly recommended to refer to amateur swimmers.
The authors understand the comment, sports nutrition guidelines do not clearly distinguish levels of practice and define recommendations based on training load.
- Before the aim of the study, the authors should report the literature gap that exist in the international literature that their study will cover.
The authors acknowledge the reviewer comment and had reinforced the introduction to justify the relevance of this research.
- At the end of the Introduction, the authors should report the aim of their study.
We thank the reviewer for point out this , it was added in lines 51-53.
- The second and the third paragraph should be enriched with data concerning amateur swimmers.
- In line 52, the authors report that their study include sub-elite swimmers. This statement should be further described.
The authors acknowledge the reviewers comment. For concordance we opted for the term amateur. These athletes were not professional but were in a competitive level in our country. They compete at regional and national events, aiming for European and world championships qualification in most cases. The term sub elite could be misunderstood therefore the authors opted for “amateur”.
- The description of the sample in lines 57-61 should be reported initially reported in the Introduction section to explain the target study population of the article.
The authors understand the comment, the introduction is now more comprehensive on the relevance of studying this athletes.
- A reference is required in line 92.
The authors thank the reviewer for pointing out this, we added a more comprehensive description and added references (lines 94-99).
- In section 2.4, the normality test that examined the continuous variables should be reported.
We acknowledge the comment ad added in line 105.
- In line 115, the p-value should be over 0.05.
The authors understand the reviewer comment but according to statistical interpretation and to our best knowledge, the p<0.05 was used to define the statistical significance.
- A more detailed description concerning the results of Tables 1, 2 and 3 should be performed, even if there are not significant differences.
We acknowledge the reviewer comment and added Lines 113-116 and lines 129-132.
- In Table 4, p-values should be added.
The p-values were added in a separate column and it is mentioned in the table caption.
- Since the swimmers of the sample is athletes, the authors should report the exact type of swimming that they are specified. This should be also analysed since each type of swimming is associated with different energy intake.
The authors understand this comment but these athletes did present yet a great degree of specificity in style. Most competed in two styles (crawl in the freestyle events and back or breast stroke). In the training sessions they did practiced all the styles in spite there was specific training for the preferred one in each competition.
- In section 4.2 the sentence in lines 168-171 is very complex and ton easily understood.
The authors agree with the reviewer coment, the sentence was rewritten (lines 144-147)
- The paragraph in lines 186-190 needs more analysis. The first sentence of this paragraph is complex and not easily understood.
We acknowledge the reviewer comment and corrected the text in lines 151-158
- Based on the Methods-Results-Discussion section, the authors could report in the title of the paper that they studied amateur athletes swimmers.
The authors agree with the reviewer, the title was revised.
- In section 4.3, the two last paragraph should be merged and should be enriched by further information.
The authors thank the reviewer for point this out and corrected the paragraph and a further explanation for this intake had been added (lines 242-243).
- In the Discussion a separate section should be added by reporting the strengths and the limitations of the study.
The authors agree with the reviewer and added the section (lines 169-278)
- The main concern of the study is the too small sample size which can not support the reported conclusions in lines 256-261.
We acknowledge the reviewer comment and revised the sentence
In the Conclusion section the authors should report what future studies could be performed in the future based on the results of their study.
We acknowledge the reviewer comment and added a comment in lines 283-285. Due to the lack of studies in this area, this can justify the interest for further research. Our research group is working in food intake and nutrition knowledge at amateur level based on these data and other that is under analysis. We appreciate the constructive comments on this work and hope
Reviewer 3 Report
Comments and Suggestions for Authors
This study aimed to evaluate energy and nutrient intake and compare with the recommend value in amateur swimmers. As the results, the amateur swimmers had shown an inadequate energy and carbohydrate intake for their activity level, and female athletes had also indicated a low intake of crucial micronutrients. The reviewer considers that the results of this study indicate nutritional problems in amateur swimmers and provide useful information from the perspective of improving athletes’ physical condition and performance. However, there are several limitations in this study.
1. What is novel about this study? There have been many previous reports on nutritional assessments of swimmers, so the reviewer cannot understand the novelty of this study. The authors should explain how the results of this study differ from those of previous studies and explain the novelty of this study.
2. Are the results of this study specific to swimmers? The reviewer thinks that this study only compared male and female athletes and that the results of present study are not specific to swimmers. The reviewer considers that comparisons with athletes from other sports are needed to show that the results of present study are specific to swimmers.
3. In relation to the above, the authors conclude that female swimmers had shown an inadequate intake of calcium, iron and zinc, and had indicated a low intake of crucial micronutrients. An insufficient energy intake has long been identified as one of the triad for female athletes. The reviewer considers that the findings of this study are not specific to female swimmers and are merely indicative of nutritional problems for general female athletes.
4. The reviewer considers that this study involved a limited population of amateur swimmers and the sample size is very small. The reviewer believes the small sample size to be one of the major limitations of this study.
5. As the authors explain in the “Introduction” section, swimmers have high energy intake requirements according to their activity levels. In this study, the authors included the adherence to Mediterranean diet in their investigations. What is the purpose of assessing adherence to the Mediterranean diet in swimmers who have high energy intake requirements? The reviewer cannot understand the purpose of assessing adherence to a Mediterranean diet in swimmers, who have high energy intake requirements.
6. In this study, the authors focused on the intake of calcium, iron, and zinc as nutrients. Why did the authors focus on calcium, iron, and zinc intakes in this study? It is unclear why the authors focused on these nutrients in this study.
7. In this study, the authors explain that swimmers are consuming insufficient energy intake relative to their activity levels. The reviewer considers that a comparison of energy intake with energy expenditure associated with swimming training and daily living is necessary to conclude that the energy intake is insufficient for the activity levels.
8. In this study, the authors compared energy and macronutrient intake between workout and rest days. However, in this article the authors did not provide any explanation for nutritional assessment methods on workout and rest days. Does the three-day food record include workout and rest days? Or is it the average of three workout days and three rest days?
9. Micronutrient results in this study are presented as absolute values. Total energy and micronutrient intake data should be presented relative value to body weight or total energy intake.
10. In addition to the mean and standard deviation, the table should also show the results of statistical processing (p-values).
Author Response
The authors would like to thank the reviewer for the constructive comments that contributed to an improvement in the manuscript.
- What is novel about this study? There have been many previous reports on nutritional assessments of swimmers, so the reviewer cannot understand the novelty of this study. The authors should explain how the results of this study differ from those of previous studies and explain the novelty of this study.
The authors acknowledge the reviewer comment and appreciate the opportunity to discuss this subject. There is in fact some research done in swimmers food intake and body composition but it is heterogenous in what refers to countries, participants age group and competitive level. Nevertheless, few studies had used methods that could be replicated in field context (skinfolds, food records). Considering that most swimmers are not professional athletes, this adds up an important variable that can affect not only habits buts also its impact in performance level. Nevertheless, the authors revised the introduction in order to justify the research gap that motivates works like this.
- Are the results of this study specific to swimmers? The reviewer thinks that this study only compared male and female athletes and that the results of present study are not specific to swimmers. The reviewer considers that comparisons with athletes from other sports are needed to show that the results of present study are specific to swimmers.
The authors understand the reviewer comment. Sports nutrition recommendations are globally addressed to several sports and there are no specific recommendations that differ from works from Shaw or the ACSM recommendations (nutrition and athletic performance). Micronutrient recommendations are defined according to the age group and gender, there are no specific micronutrient requirements for athletes at our best knowledge.
- In relation to the above, the authors conclude that female swimmers had shown an inadequate intake of calcium, iron and zinc, and had indicated a low intake of crucial micronutrients. An insufficient energy intake has long been identified as one of the triad for female athletes. The reviewer considers that the findings of this study are not specific to female swimmers and are merely indicative of nutritional problems for general female athletes.
The authors acknowledge and appreciate the reviewer comment, in fact as presented in lines 217 and 239. Other studies reported this inadequate intake in swimmers including both genders. For zinc, we also added some other results in line 246.
- The reviewer considers that this study involved a limited population of amateur swimmers and the sample size is very small. The reviewer believes the small sample size to be one of the major limitations of this study.
The authors acknowledge the reviewer comment and add limitations at the end of the discussion section.
- As the authors explain in the “Introduction” section, swimmers have high energy intake requirements according to their activity levels. In this study, the authors included the adherence to Mediterranean diet in their investigations. What is the purpose of assessing adherence to the Mediterranean diet in swimmers who have high energy intake requirements? The reviewer cannot understand the purpose of assessing adherence to a Mediterranean diet in swimmers, who have high energy intake requirements.
The authors acknowledge and appreciate the reviewer comment, this was added to the revised version of the manuscript introduction.
- In this study, the authors focused on the intake of calcium, iron, and zinc as nutrients. Why did the authors focus on calcium, iron, and zinc intakes in this study? It is unclear why the authors focused on these nutrients in this study.
The authors acknowledge and appreciate the reviewer comment, this was added to the revised version of the manuscript introduction.
- In this study, the authors explain that swimmers are consuming insufficient energy intake relative to their activity levels. The reviewer considers that a comparison of energy intake with energy expenditure associated with swimming training and daily living is necessary to conclude that the energy intake is insufficient for the activity levels.
The authors thank the reviewer for pointing out this topic. Due to limitations in controlling the energy expenditure during training sessions, the research team had collected data on training frequency, session duration and some technical data on expected effort. In addition to that, the initial assessment included some questions on daily routine habits in order to establish the activity level. The authors added this explanation in lines 96-99.
- In this study, the authors compared energy and macronutrient intake between workout and rest days. However, in this article the authors did not provide any explanation for nutritional assessment methods on workout and rest days. Does the three-day food record include workout and rest days? Or is it the average of three workout days and three rest days?
The authors acknowledge the reviewer comment and add the explanation on the food record days in line 86, as well as its analysis in line 108. The tables refer to average training day food intake as well as average intake in 3 days, because there were no differences in the energy intake within the three days.
- Micronutrient results in this study are presented as absolute values. Total energy and micronutrient intake data should be presented relative value to body weight or total energy intake.
The authors understand the reviewer comment and added a supplementary file (Figure S1) commented in Lines 139-140. In sports nutrition, the macronutrient contribution to energy intake is commonly replaced by recommendations in grams per kg as presented in Table 3.
- In addition to the mean and standard deviation, the table should also show the results of statistical processing (p-values).
The authors acknowledge the reviewer comment and added the relevant p-values in tables 2, 3 and 4.
Round 2
Reviewer 2 Report
Comments and Suggestions for Authors
The authors have significantly improved their manuscript after the revision process.
Reviewer 3 Report
Comments and Suggestions for Authors
I think all responses to reviewers' comments have been addressed satisfactorily.
I have no comments on the revised manuscript.